# Differing taxonomic responses of mosquito vectors to anthropogenic land-use change in Latin America and the Caribbean

**Isabel K. Fletcher** [1,2]*, **Rory Gibb** [1,2,3], **Rachel Lowe** [1,2,4,5], **Kate E. Jones** [3]

**1** Centre for Mathematical Modelling of Infectious Diseases, London School of Hygiene & Tropical Medicine, London, United Kingdom, **2** Centre on Climate Change and Planetary Health, London School of Hygiene & Tropical Medicine, London, United Kingdom, **3** Centre for Biodiversity and Environment Research, University College London, London, United Kingdom, **4** Barcelona Supercomputing Center (BSC), Barcelona, Spain, **5** Catalan Institution for Research and Advanced Studies (ICREA), Barcelona, Spain

\* isabel.fletcher@lshtm.ac.uk

**Data Availability Statement:** All data and code that support the findings of this study are available in the Github repository https://github.com/isabelfletcher/vectors_landuse.

## Abstract

Anthropogenic land-use change, such as deforestation and urban development, can affect the emergence and re-emergence of mosquito-borne diseases, e.g., dengue and malaria, by creating more favourable vector habitats. There has been a limited assessment of how mosquito vectors respond to land-use changes, including differential species responses, and the dynamic nature of these responses. Improved understanding could help design effective disease control strategies. We compiled an extensive dataset of 10,244 *Aedes* and *Anopheles* mosquito abundance records across multiple land-use types at 632 sites in Latin America and the Caribbean. Using a Bayesian mixed effects modelling framework to account for between-study differences, we compared spatial differences in the abundance and species richness of mosquitoes across multiple land-use types, including agricultural and urban areas. Overall, we found that mosquito responses to anthropogenic land-use change were highly inconsistent, with pronounced responses observed at the genus- and species levels. There were strong declines in *Aedes* (-26%) and *Anopheles* (-35%) species richness in urban areas, however certain species such as *Aedes aegypti*, thrived in response to anthropogenic disturbance. When abundance records were coupled with remotely sensed forest loss data, we detected a strong positive response of dominant and secondary malaria vectors to recent deforestation. This highlights the importance of the temporal dynamics of land-use change in driving disease risk and the value of large synthetic datasets for understanding changing disease risk with environmental change.

## Author summary

An understanding of the response of disease vectors to anthropogenic activities can aid in the control of mosquito-borne disease transmission. However, regional assessments of these responses are lacking, especially in areas where mosquito-borne diseases are emerging and re-emerging. We assembled a synthetic dataset of mosquito abundance and

**Funding:** IKF was supported by the Biological and Biotechnology Research Council (grant BB/M009513/1) and RL was supported by a Royal Society Dorothy Hodgkin fellowship. RG was supported by a Graduate Research Fellowship from University College London. The funders had no role in study design, data collection and analysis, decision to publish, or preparation of the manuscript.

**Competing interests:** The authors have declared that no competing interests exist.

species richness across multiple land-use types in Latin America and the Caribbean, using data from peer-reviewed published studies. We then used this dataset to explore whether mosquito abundance and species richness changed, compared to a baseline of minimal anthropogenic activity, depending on the type of land use. Overall, we observed a decline in mosquito biodiversity in urban areas. We detected distinct species-specific responses in abundance to land use, with some important disease vectors, such as *Aedes aegypti*, increasing in abundance in anthropogenic environments. Finally, we also demonstrated by coupling our dataset with forest loss data that the abundance of dominant malaria vectors in the region increases with deforestation.

## Introduction

The global land system is facing mounting pressure from anthropogenic activities, including the conversion of natural environments for agricultural practices and urban development [1]. Globally, 75% of the land surface area has been transformed by anthropogenic activities, mostly through a global net loss of forest cover and the expansion of global agriculture [2,3]. The disruption of ecosystems has devastating consequences for global biodiversity [4] and similarly influences the incidence and emergence of infectious diseases [5–8]. An improved understanding of how important disease vectors are impacted by land-use alterations is essential given current trends in land-use transformation and climate change [9], as well as the emergence and re-emergence of infectious diseases [10,11].

Mosquito-borne diseases are particularly sensitive to ecological alterations resulting from land-use transformations, including changes in vector habitat availability and vector-human contact rates [12]. For example, ecological changes caused by deforestation due to agricultural development in the Brazilian Amazon increase the abundance and biting of the principal malaria vector, *Anopheles darlingi* [13,14]. At intermediate levels of deforestation in agricultural frontier regions, greater amounts of forest edge habitat provide suitable conditions for the proliferation of *An. darlingi* mosquitoes. Subsequently, this elevates malaria risk only in the early stages of land-use alterations when the amount of forest edge habitat is at its highest [15–17]. In addition to facilitating increases in habitat suitability for mosquito vectors, land-use change, such as agricultural development, also increases human exposure to pathogen-carrying mosquitoes [18]. These local-scale studies have demonstrated how land-use changes can alter disease risk through modification of vector habitats. However, there is limited understanding of whether mosquito responses to land-use change are consistent at a regional scale. An assessment of how important vectors respond to land-use change and dynamic ecological alterations, such as deforestation, will be useful for designing control strategies that can be implemented at scale.

Given the distinct life history characteristics and diversity of mosquito species (over 3,600 recognised Culicidae species [19]), it is likely that species will respond differently to land-use change. Urbanisation negatively impacts native terrestrial biodiversity [4] and allows for synanthropic mosquitoes, which live in or near human dwellings, to persist in novel environments [20]. This is due to the diverse range of aquatic habitats for mosquito breeding that exists in urban environments, such as water-storage containers, discarded plastic, and drains [21]. Increased provisioning of vector habitats, in addition to the availability of human hosts, has enabled synanthropic mosquitoes, such as the dengue vector *Ae. aegypti* and malaria vector *An. stephensi*, to flourish in urban environments [22–25]. In contrast, mosquito biodiversity is higher in rural, forested landscapes [26,27], with some mosquitoes exhibiting a preference for

preserved forested habitats [28,29]. Despite this understanding of species' habitat preferences, there is a limited understanding of whether mosquito species respond differentially to anthropogenic land-use change. In addition, several local-level studies have provided an increased mechanistic understanding of how habitat alterations such as deforestation, favour important disease vectors [13]. However, there has been a limited assessment of whether consistent responses to deforestation can be detected regionally and how the dynamic nature of these responses compares among mosquito species. Such assessments will be useful for developing effective mosquito control strategies that can be tailored to species behaviour, such as feeding and resting patterns.

In Latin America and the Caribbean region (LAC) mosquito-borne diseases are a dynamic public health threat. Approximately five million dengue cases were reported in LAC in 2020 [30], and 145 million people in the Americas are at risk of malaria and cases of yellow fever occurring in 13 countries across the region, including Peru, Bolivia and Brazil [31]. Globally, the Amazon rainforest serves as the largest reservoir of arboviruses [32] and is subject to intensifying human pressures, including the development of land for pasture and deforestation for soybean production [2,33]. The alterations to natural landscapes have resulted in the rapid expansion of mosquito-borne diseases, such as dengue and yellow fever [34,35], as well as the rapid re-emergence of malaria in Venezuela [36].

Here, we developed a regional approach to assess the response of *Aedes* spp. and *Anopheles* spp. mosquitoes to land-use change across LAC. We employed a systematic data search strategy to compile an extensive dataset of mosquito abundance records and used a comparative space-for-time approach to identify taxonomic responses to anthropogenic land-use change. We tested for spatial differences in mosquito abundance responses to anthropogenic land-use change and examined whether mosquito species richness in human-dominated landscapes is reduced compared to areas minimally affected by human activity. Additionally, we investigated the temporal dynamics of land-use change by testing for differences in mosquito species responses to recent deforestation, harmonising findings from local level studies.

## Methods

### Mosquito biodiversity dataset construction and assignment of land-use categories

A dataset of *Aedes* and *Anopheles* mosquito species in LAC across multiple land-use types was built by extracting relevant abundance data from published studies using a systematic data search strategy (S1 Text, S1 Fig and S2 Table). The construction of the dataset followed the methodology in Hudson *et al.* [37] for the PREDICTS database (a global compilation of site-level ecological data across different land uses and land-use intensities). Species- and site-specific abundance data were extracted for each included study, and information on the sampling methodology, the study area and site descriptions were collected (S2 Table). As with the PREDICTS database, each study site was nested to account for variation due to sampling methodology. Specifically, each record was assigned a study number (a unique paper), site number (a geographic location at which mosquito abundance was sampled), study block (a collection of sites within a distinct spatial cluster, to account for spatial autocorrelation within a study) and study sample (a sample with consistent sampling methodology, such as capture method and sample month) (S2 Table).

Each sample site was assigned a land-use type (primary vegetation, secondary vegetation, managed or urban) and use intensity (minimal or substantial), modified from criteria developed by Hudson *et al.* [37] and Gibb *et al.* [8] (S3 Table). Sites were labelled based on the predominant land-use type described in each study's site description, and the use intensity was

assigned based on the degree of human activity at each site. For example, sites sampled near or within small rural villages, biological reserves, research stations or forested areas, were labelled as primary vegetation with minimal or substantial use. Managed sites included plantations, pastures or croplands [8]. Urban sites were characterised by the presence of paved roads and significant impervious surface area. For analysis, land-use type and intensity were combined into a categorical variable. Minimal and substantial use intensities were retained for primary vegetation sites and due to a lack of data representation, use intensities for secondary vegetation, managed and urban sites were combined into a single category. This resulted in a categorical variable with five levels; primary vegetation-minimal, primary vegetation-substantial, secondary vegetation, managed and urban (S4 Table).

## Modelling the effects of land use on mosquito abundance and species richness

Bayesian mixed-effects models were developed to assess the spatial variation of mosquito biodiversity (species-level abundance and site-level species richness) across different land-use types (S5 Table). In studies where sampling effort varied across sample sites, raw species-level abundance measurements were divided by sampling effort to obtain effort-corrected abundance measurements [4,38,39]. Due to the high number of zero observations, site-level species abundance measurements were overdispersed. To address this, abundance measurements were log-transformed and subsequently modelled using a Gaussian likelihood. Site-level species richness (the number of uniquely named species sampled at each site) was modelled using a Poisson likelihood [4,8]. Models were constructed to analyse the abundance of *Aedes* species, *Anopheles* species and both species combined. Species-specific responses in abundance to land use were also examined, by building separate models for four mosquito species per genus. The selection of mosquito species was based on their representation in the dataset, with priority given to the species with the highest number of records (S6 Table), whilst ensuring the selected species are vectors of at least one human disease [40,41]. To avoid confounding factors related to mosquito habitat preferences and ranges, the models for each species only included studies where the respective species was detected.

All models included a random intercept term for each study to account for variation between studies, including reporting methods and sampling methodologies (i,e. outdoor vs. indoor sampling and trapping method). A random effect for each study site was included to account for overdispersion due to site-level differences [37]. Abundance models also included a random intercept for each unique species, resulting in multiple observations per site. This allowed for species-specific variation in abundance that could result from differences in feeding and resting behaviour, meaning some species were easier to sample than others. Other random effects considered in the model structure included study block, used to account for spatial autocorrelation between sites within a study and study sample. Ecoregion, reflecting habitat types of terrestrial ecoregions of the world [42] was also included as a random effect to account for the confounding effects of climate and habitat [37]. The best-fitting random effects structure was determined by formulating iterative models of each response variable (i.e. abundance and richness) with the addition of each random effect (S7 and S8 Tables).

To assess model adequacy, Bayesian metrics the deviance information criterion (DIC) [43] and the Watanabe-Akaike information criterion (WAIC) were used [44]. The inclusion of random effects in the final model was determined based on improvements in model fit. This was assessed by a reduction in DIC and WAIC with the addition of each random effect, although ecoregion was retained in all models to avoid the potential confounding influence of climate and habitat. Model fit was also assessed visually by examining the agreement between fitted

versus observed values (S2 Fig). All models were implemented in a Bayesian framework, using R-INLA [45].

Species richness and abundance models were cross-validated by testing the sensitivity of the fixed effects estimates to geographical and random subsampling. For geographical subsampling, models were fitted by excluding data from Brazil, where data coverage was highest. Models were also fitted to data excluding each ecoregion (n = 6) at a time. Finally, for the random subsampling, eight hold-out models were fitted, where each model excluded 12.5% of randomly selected samples of the data at a time.

### Modelling the impact of deforestation on mosquito biodiversity

To explore the temporal dimension of land-use change, we compared species-specific mosquito responses to deforestation by combining *Aedes* and *Anopheles* abundance records from primary and secondary vegetation sites, with remotely-sensed deforestation data [46]. Abundance records were combined with deforestation data obtained from the Hansen dataset, which provides spatially continuous annual estimates of forest loss derived from Landsat images, between 2000–2019 [46]. For each unique primary and secondary vegetation site in our dataset, we extracted the percentage of forest loss within a 320 m buffer around each site. A 320 m buffer was used as an approximation of mosquito flight distance, which can range between 50 m and 50 km. Average *Aedes* and *Anopheles* flight distances range between 89–542 m [47] so the mean of these values was used. As the time since deforestation greatly influences mosquito dynamics and subsequent disease risk [16], deforestation data was temporally matched with site-level mosquito abundance and richness records. An estimate of recent forest loss was obtained by using estimates from the last five years since the sampling start date at each site. Bayesian mixed-effects models for *Aedes* and *Anopheles* species richness and abundance were formulated including site-level proportional deforestation as a linear covariate. As with the land-use models, random effects for study number, site number, study sample and terrestrial ecoregion were also included (S9 Table). Eight species-specific abundance models selected based on data representation were also formulated to test for individual responses to deforestation.

## Results

### Dataset of mosquito biodiversity and land use

The final mosquito abundance dataset comprised 10,244 records collected from 632 sites and obtained from 93 studies that were identified in the systematic data search (Fig 1A). Most sampled sites were primary vegetation (46%, n = 292; Fig 1A), which represented 37% of total records in the dataset (n = 3,835). The dataset covered 13 countries across the LAC region, including Mexico, French Guiana, Argentina, Colombia and Venezuela (Figs 1A and S3 and S10 Table) and coverage was highest in Brazil (67% of records, n = 6,870; S10 Table), and in biodiversity hotspots such as the Amazon basin (68% of total sites, n = 431; Fig 1B) and Atlantic Forest (18% of total sites, n = 111; Fig 1A). The dataset spanned six terrestrial ecoregions (S6 Fig), the majority of which were in the Amazon and were forested ecoregions (96% of total sites, n = 609; Fig 1B). The dataset included 91 species (S11 Table), of which 36% (n = 33) were *Aedes* species and 64% (n = 58) were *Anopheles* species (Fig 1C).

### Effect of land use on mosquito species richness and abundance

We found limited evidence of consistent effects of land use on mosquito species richness, except in urban areas (Figs 2A and S4A and S12 Table). *Aedes* species richness was reduced by

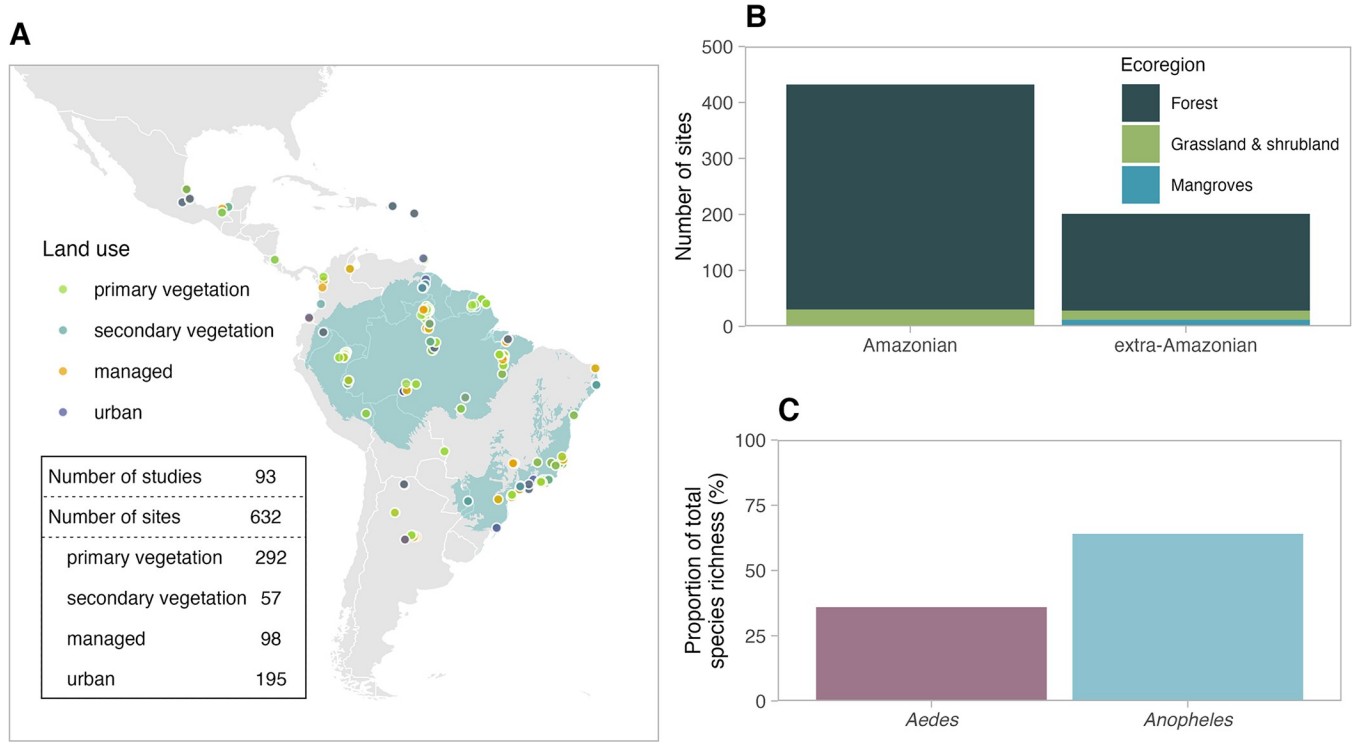

**Fig 1. Dataset of *Aedes* and *Anopheles* mosquito biodiversity in Latin America and the Caribbean.** Geographical location (points) of surveyed sites (n = 632) and their predominant land-use type across 93 collated studies (A). Colours represent the four land-use types: primary vegetation (green), secondary vegetation (blue), managed (orange) and urban (purple). Green shading on the map shows the Amazon basin, base map obtained from Harvard WorldMap [48] and Atlantic Forest, map obtained from Muylaert *et al.* [49]. The number of surveyed sites across broadly defined terrestrial ecoregions (forests, grassland and shrubland, and mangroves) are shown for Amazonian and extra-Amazonian regions (the remaining LAC region) (B). Proportion (%) of unique species (species richness) across total species richness in the dataset (C). Base map sourced from rnaturalearth [50].

26% in urban areas, compared to the primary vegetation minimal use baseline (95% CI: -42.7% to -5.2%; Fig 2A). *Anopheles* mosquito species richness demonstrated a larger 35% decline in richness in urban areas (95% CI: -49.8% to -14.3%). In managed areas, there was a trend towards increased *Anopheles* richness, although the credible intervals crossed zero indicating uncertainty (Fig 2A). Relative to primary vegetation, *Anopheles* mosquitoes in urban sites experienced a significant 13% (95% CI: -22.4% to -2.1%) reduction in abundance and there was also a trend towards decreased *Aedes* abundance in urban sites (Fig 2B and S13 Table). In contrast, abundance of *Anopheles* mosquitoes increased by 11% in managed sites (95% CI: 0.2% to 24.0%). There was a minimal effect of other land-use types on overall *Aedes* and *Anopheles* abundance. Total *Aedes* and *Anopheles* mosquito species richness was 38% lower in urban landscapes (95% CI: -47.9% to -26.8%), and there was no significant effect of land-use type on total mosquito abundance (S4B Fig).

*Aedes* and *Anopheles* species richness and abundance models were broadly robust to geographical subsampling, although there were higher levels of uncertainty in abundance and richness estimates when data from Brazil were excluded from the models (S5 Fig). Urban estimates were particularly sensitive to exclusion of Brazilian data, likely due to the high number of urban sites in Brazil (Fig 1A and S10 Table). We also found that *Aedes and Anopheles* species richness models were highly influenced by sites from tropical rainforests, highlighting the need for more representative sampling outside this ecoregion (S6 Fig). Finally, abundance and species richness responses were largely robust to random subsampling (S7 Fig).

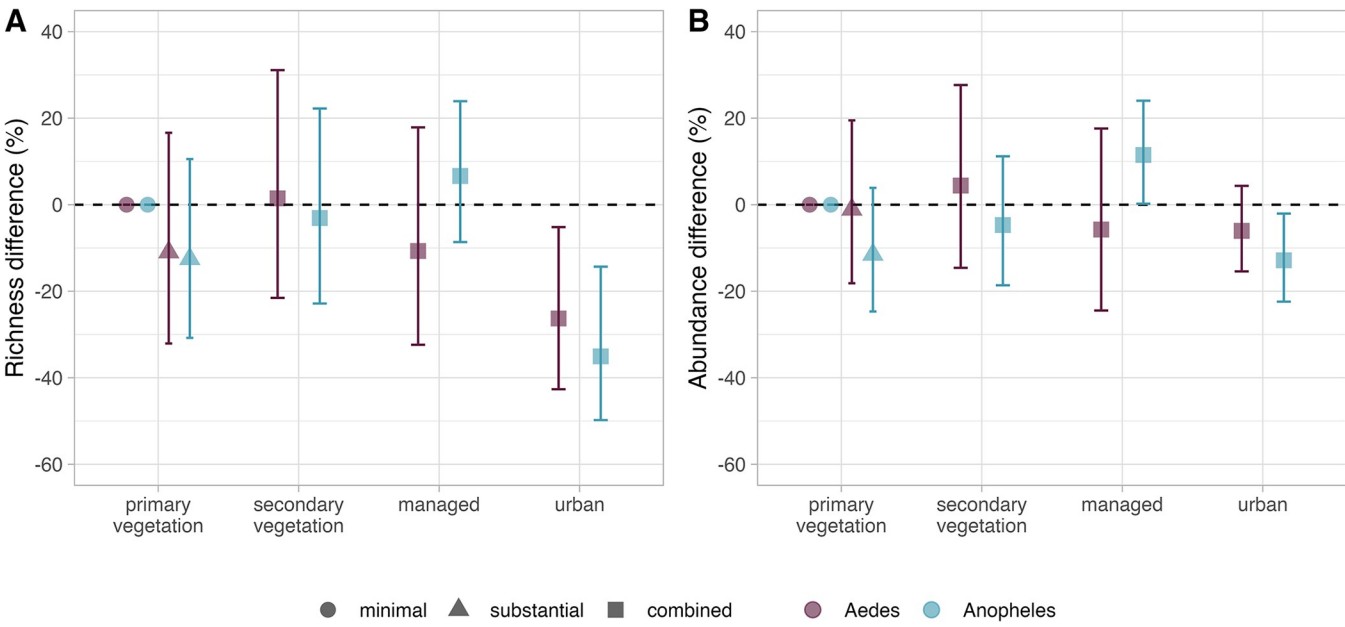

**Fig 2. Responses of mosquito species richness and abundance to land-use type and intensity.** *Aedes* (purple) and *Anopheles* (blue) mosquito species richness (A) and abundance (B) responses to land-use types with minimal (circles), substantial (triangles) and combined (squares) use intensities. Effect sizes were adjusted to a percentage by expressing each mean fixed effect and 95% credible intervals as a percentage of the baseline (primary vegetation minimal use, shown as zero). Intensity levels for secondary vegetation, managed and urban land uses were aggregated due to a lack of data representation.

## Species-specific mosquito abundance responses to land use

We investigated species-specific variation in responses to land use, by analysing the mean effects of land use on species-level abundance for four *Aedes* and four *Anopheles* mosquito species that were most represented in the dataset (S6 Table). We found that there was a high degree of divergence in the abundance responses of mosquito species to land use (Fig 3 and S14 Table). *Ae. aegypti* and *Ae. albopictus* exhibited contrasting responses to substantial use intensity at both primary and secondary vegetation sites (Fig 3). *Ae. aegypti* demonstrated a negative abundance response of 56% (95% CI: -75.8% to -22.4%) at secondary vegetation sites and a 42% decline at primary vegetation sites with substantial use (95% CI: -65.2% to -3.4%). In contrast, *Ae. albopictus* showed elevated abundance at both substantial use primary vegetation (95%; 95% CI: 26.3%– 198.9%) and secondary vegetation sites (68%; 95% CI: 4.9% - 167.3%). Among the eight species analysed, *Ae. aegypti* demonstrated the largest abundance, with a 195% increase in abundance at managed sites (95% CI: 59.1% - 446.8%), although there was a high degree of uncertainty associated with this estimate. Both *Ae. aegypti* and *Ae. albopictus* demonstrated a trend of increased abundance at urban sites, although this was not significant. In contrast to *Ae. albopictus*, *Ae. scapularis* demonstrated a 44% reduction in abundance at primary vegetation sites with substantial use intensity (95% CI: -57.5% to -27.1%). Similarly, *Ae. serratus* abundance was reduced by 61% at primary vegetation sites with substantial use (95% CI: -75.5% to -36.6%) and by 66% in managed sites (95% CI: -79.3% to -44.5%).

The response of *Anopheles* mosquito abundance to different land uses, in contrast to *Aedes*, was less marked (Fig 3). Among the *Anopheles* species analysed, only the abundance of *An. albitarsis* was altered in comparison to the primary vegetation baseline. At managed sites, *An. albitarsis* abundance was 163% higher (95% CI: 34.6% - 422.2%). We detected a minimal impact of land use on *An. albimanus* and *An. nuneztovari* abundance and although the

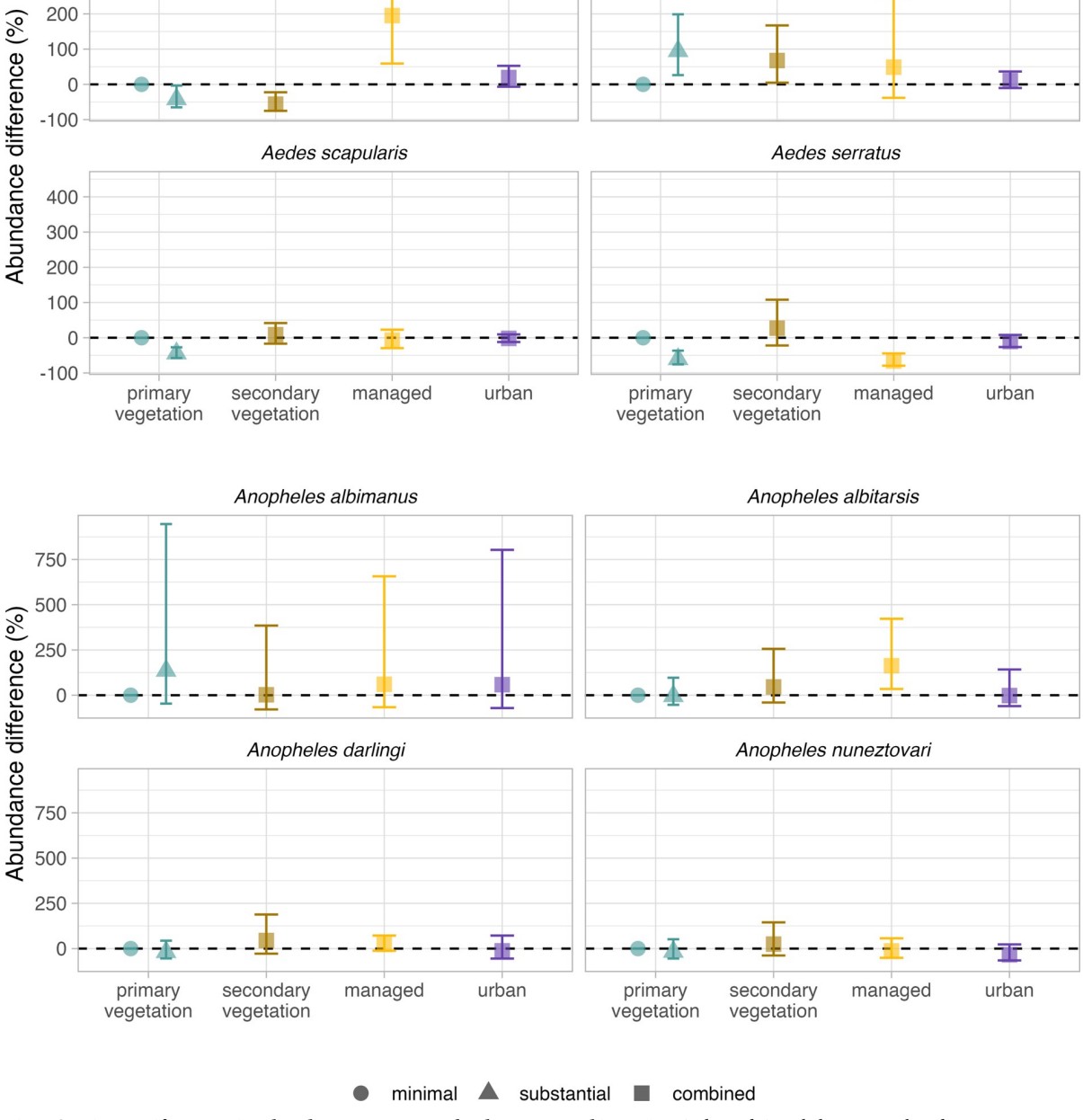

**Fig 3. Species-specific mosquito abundance responses to land-use type and intensity.** *Aedes* and *Anopheles* species abundance responses to land-use types with minimal (circles), substantial (triangles) and combined (squares) use intensities. For each genus, the four most represented species in the dataset were selected. Effect sizes were adjusted to a percentage by expressing each mean fixed effect and 95% credible intervals as a percentage of the baseline (primary vegetation minimal use, shown as zero). Intensity levels for secondary vegetation, managed and urban land uses were aggregated due to a lack of data representation.

credible intervals crossed zero, there was evidence of a trend towards higher *An. darlingi* abundance at secondary vegetation sites. There was a high level of uncertainty in the estimates for *An. albimanus*, possibly due to sparse sampling (only three urban, four managed, five secondary vegetation and ten primary vegetation sites; S15 Table). When influential mosquito species

records were held out from genus-level abundance models, the overall response to land use did not change markedly (S8 Fig). However, models excluding *Ae. albopictus* records were sensitive to exclusion of data, as were estimates for managed land-use types.

### Influence of deforestation on mosquito biodiversity

We observed a strong impact of deforestation on malaria-transmitting mosquito species. Recent deforestation, in the last five years, was associated with higher *Anopheles* species richness (mean estimate 0.13, 95% CI: 0.03–0.23; Fig 4A). This result corresponds to a 14%

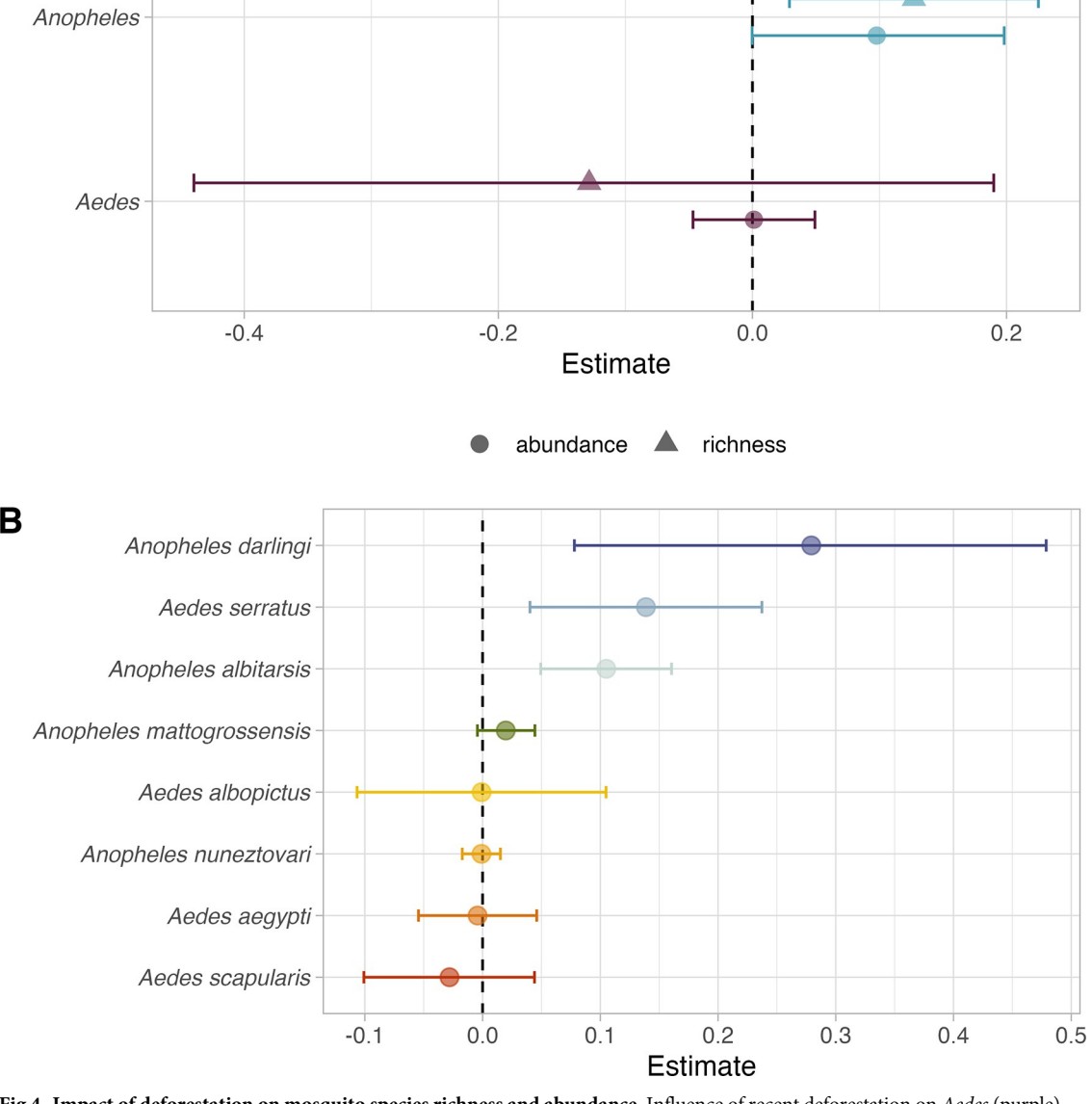

**Fig 4. Impact of deforestation on mosquito species richness and abundance.** Influence of recent deforestation on *Aedes* (purple) and *Anopheles* (blue) mosquito abundance (circles) and species richness (triangles; A). Abundance responses of eight mosquito species to recent deforestation (in the last five years; B). Points and bars for A and B show posterior mean and 95% credible intervals for linear fixed effects estimates of recent deforestation, calculated as proportional forest loss within the last five years of the sample start date for each site-level record.

  

increase in richness with every 1% increase in forest loss. Although not significant, there was also a trend towards increased abundance of *Anopheles* mosquitoes (Fig 4A). In contrast, we detected a minimal impact of deforestation on *Aedes* mosquito abundance and species richness. Furthermore, we found evidence of species-specific responses to deforestation. Whilst two *Anopheles* species exhibited positive responses to deforestation, there was a minimal impact of deforestation on the abundance of *Aedes* species (Fig 4B). *An. darlingi* demonstrated the largest increase in abundance with deforestation (mean estimate 0.28, 95% CI: 0.08–0.48; Fig 4B), followed by *An. albitarsis* (mean estimate 0.11, 95% CI: 0.05–0.16; Fig 4B). This corresponds to a 32% increase in *An. darlingi* abundance and a 12% increase for *An. albitarsis* with every 1% unit increase in forest loss. *Ae. serratus* also demonstrated a 15% abundance increase with forest loss (mean estimate 0.14, 95% CI: 0.04–0.24).

## Discussion

Our space-for-time approach has provided an enhanced understanding of general responses of *Aedes* and *Anopheles* mosquitoes to land-use change. Overall, we observed inconsistent spatial responses of mosquito biodiversity to land use in Latin America and the Caribbean, but a strong response to recent deforestation. Both *Aedes* and *Anopheles* mosquito species richness were reduced in urban environments, while the abundance of several synanthropic arbovirus vectors (*Ae. aegypti* and *Ae. albopictus*) and a secondary malaria (*An. albitarsis*) vector was greater in human-dominated landscapes. Further, we found a strong and consistent temporal signal of deforestation on both dominant and secondary malaria vectors, highlighting the importance of considering the temporal dynamics of land-use change in assessing disease risk. By integrating local landscape-level mosquito abundance records across 632 sites, we detected substantial taxonomic differences in biodiversity responses to land use, providing a clear proof of concept for this methodology. With the inclusion of more data, these methods could be extended to larger regional and global scales to investigate the spatiotemporal dynamics of disease vector responses to environments influenced by anthropogenic activities.

Land-use change is expected to lead to an overall decline in biodiversity, primarily due to habitat loss. However, disturbance can also favour opportunistic species that are able to adapt to and thrive in anthropogenic environments [20,51,52]. The strong decrease in *Aedes* (26% reduction) and *Anopheles* (35% reduction) mosquito species richness in urban areas in this study aligns with previous research demonstrating reduced mosquito biodiversity in urban and fragmented landscapes [53,54]. In some instances, biodiversity can provide a protective effect for disease emergence by regulating the abundance of vectors through intra- and inter-species competition, as well as through predation [55]. Disruption of this protective effect can facilitate increased abundance of specific species capable of adapting to novel environments. For example, decreased mosquito biodiversity in agricultural frontiers in the Amazon favours higher abundances of *An. darlingi* and drives subsequent malaria risk [56]. Similarly, in a malaria endemic region of Colombia, communities of *Anopheles* mosquitoes were less diverse in highly fragmented landscapes compared to more intact landscapes [54]. In addition, we also found that deforestation was associated with increased *Anopheles* species richness, suggesting that the ecological changes resulting from deforestation create novel habitats that support the proliferation of certain mosquito species [12,13].

In addition to genus-specific responses to land-use change, we found pronounced species-specific abundance responses, highlighting the complexity in generalising mosquito responses to understand disease risk. The uncertainty in mosquito responses to land use further suggests that spatial effects are highly variable over a large geographical area. The differential response of mosquito species to land-use change is likely to be driven by the unique life-history

characteristics and habitat preferences of each species [57,58]. We found increased abundance of opportunistic species in disturbed landscapes, including *Ae. aegypti* and *Ae. albopictus* mosquitoes. Human-dominated landscapes provide a range of novel habitats that facilitate increased abundance, densities, development and survival of *Aedes* mosquitoes in urban and agricultural areas [59,60]. These findings have implications for the emergence of arboviruses transmitted by *Ae. aegypti* and *Ae. albopictus*, such as dengue, yellow fever and chikungunya, as disease transmission could be facilitated in anthropogenic environments without the implementation of adequate control measures.

*An. darlingi* is a highly efficient anthropophilic malaria vector that predominates in the Amazon region [40,61]. It has been well-documented that *An. darlingi* exhibits a preference for disturbed deforested landscapes, especially in locations close to human settlements in agricultural frontier regions [13,15]. Here, *An. darlingi* exhibited the strongest response to recent deforestation, however there was high uncertainty in the spatial effects of land use. This result is supported by previous findings demonstrating that *An. darlingi* thrives in deforested areas [12,13]. Secondary growth, particularly at forest fringes created during the earlier and more rapid stages of deforestation, offers a range of suitable environmental conditions for *An. darlingi*, including increased sunlight, refugia and ground pools [15,62,63]. Our findings suggest that the temporal effects of land-use change, specifically deforestation on *An. darlingi*, have a more pronounced impact compared to the uncertain spatial effects. The relationship between deforestation and mosquito-borne disease risk is inherently complex and may be dependent on fine-scale factors, such as microclimatic variation and predation [64,65], which cannot easily be generalised across broad geographical scales.

Despite providing evidence of species- and genus-specific responses to land-use change, this study has several limitations. First, owing to the high levels of mosquito biodiversity in the Amazon and Atlantic forests captured in this study, the dataset is geographically biased towards these regions and rainforest biomes. However, the findings of this study remained robust to both random and species-level subsampling. Second, studies included in the dataset may have underestimated the true abundance of mosquito species. The sampling methods employed in each study were likely biased towards anthropophilic mosquitoes and species that are easier to find and capture. Nonetheless, several studies included in the dataset sampled mosquitoes using multiple sampling methods. For example, mosquito sampling was performed in many studies using human-landing catches, which primarily captures anthropophilic mosquitoes, although can be used to capture both endophilic (indoor-resting) and exophilic (outdoor) mosquitoes [66]. Other studies used baited traps and ovitraps left overnight to capture nocturnal mosquitoes and those at different life stages, such as larvae and pupae.

In addition, the random effects structure employed in the models accounted for differing sampling methodologies across studies. This approach helps account for a proportion of the variation in mosquito abundance observed. The mosquito species included in the dataset are likely to be biased towards dominant and incriminated vector species, such as *An. darlingi* and *Ae. aegypti*. Future assessments could consider species bias by taking into account publication effort [8] and where possible, ensuring the inclusion of under-represented species that may well be efficient vectors of human diseases. Additionally, abundance records in the dataset included a substantial number of zero observations, resulting from species sampling at sites where occurrence was low. Incorporating species occurrence probability into the modelling framework may be a method to address the zero-inflation of abundance data [8].

Ecological changes caused by anthropogenic land-use change have a wide range of cascading effects on mosquito-borne disease risk. A comprehensive grasp of how mosquito species are affected by anthropogenic disturbance will facilitate the development of highly effective

disease control measures. A greater understanding could additionally equip vector control efforts with species-specific information to support targeted elimination efforts for mosquito-borne diseases such as dengue, yellow fever, malaria and chikungunya. This study has presented a comparative dataset of 10,244 *Aedes* and *Anopheles* mosquito records in Latin America and the Caribbean, which is a valuable resource for investigating the effect of land-use change on mosquito-borne disease risk that is epidemiologically relevant at the regional scale. We demonstrate considerable species-specific responses, which represent the diverging impacts of land-use change on mosquito fauna and caution against generalising predictions of vector responses to environmental change. These findings strengthen our understanding of how opportunistic species contribute to mosquito-borne disease risk in anthropogenic environments.

## Supporting information

**S1 Text. Systematic data search strategy.** Systematic search strategy used to find and extract mosquito abundance data, including inclusion and exclusion criteria.
(DOCX)

**S1 Table. Search terms used for systematic data search.** Mosquito, geographical and land use specific terms used to systematically search three databases (Medline, Scopus and Web of Science) for *Aedes* and *Anopheles* mosquito abundance records across multiple land-use types in Latin America and the Caribbean. * denotes wildcard terms.
(DOCX)

**S2 Table. Summary of site-level information extracted from included studies.** Site-level information extracted from each included study to formulate a dataset of mosquito vector biodiversity over different land-use types. The nested structure of the dataset (study number, site number, study block and study sample) followed that of the PREDICTS database (37).
(DOCX)

**S3 Table. Land-use categories used in the dataset.** Description of land-use types used to classify sample sites in the dataset. Categories were adapted following Hudson *et al.* (37) and Gibb *et al.* (8).
(DOCX)

**S4 Table. Land-use intensity categories used in *Aedes* and *Anopheles* abundance and species richness models.** Site-level distribution and number of abundance records per land-use category. Use intensity for managed and urban land-use types were aggregated due to low data representation.
(DOCX)

**S5 Table. Summary of land-use intensity models.** Summary of components of total, *Aedes* and *Anopheles* abundance and species richness models. The number of sites and site-level records in each model is shown.
(DOCX)

**S6 Table. *Anopheles* and *Aedes* mosquito species included in species-specific abundance models of land-use intensity.** Site-level distribution and number of site-level abundance records per *Aedes* and *Anopheles* species with greatest representation in the dataset.
(DOCX)

**S7 Table. Iterative models for selecting the best-fitting random effects structure for models of land-use intensity and species richness.** Deviance information criterion (DIC) and

Watanabe-Akaike information criterion (WAIC) for models of total, *Aedes* and *Anopheles* species richness with the addition of random effects structures. Each random effect was added iteratively to assess model performance.
(DOCX)

**S8 Table. Iterative models for selecting the best-fitting random effects structure for models of land-use intensity and abundance.** Deviance information criterion (DIC) and Watanabe-Akaike information criterion (WAIC) for models of total, *Aedes* and *Anopheles* abundance with the addition of random effects structures. Each random effect was added iteratively to assess model performance.
(DOCX)

**S9 Table. Summary of deforestation models.** Summary of components of total, *Aedes* and *Anopheles* abundance and species richness models in response to recent deforestation. The number of sites and site-level records in each model is shown. Only records at primary and secondary vegetation sites were included.
(DOCX)

**S10 Table. Site distribution by country and land-use type.** Number of sites included in the dataset by country and land-use type.
(DOCX)

**S11 Table. List of *Aedes* and *Anopheles* mosquito species included in abundance and species richness models.** List of *Aedes* and *Anopheles* mosquitoes (n = 91) included in models and number of abundance records per species.
(DOCX)

**S12 Table. Parameter estimates for land-use types in mosquito species richness models.** Posterior mean estimates, lower (2.5%) and upper (97.5%) credible intervals (CI) for land-use types in species richness models of *Aedes* and *Anopheles* mosquitoes.
(DOCX)

**S13 Table. Parameter estimates for land-use types in mosquito abundance models.** Posterior mean estimates, lower (2.5%) and upper (97.5%) credible intervals (CI) for land-use types in abundance models of *Aedes* and *Anopheles* mosquitoes.
(DOCX)

**S14 Table. Parameter estimates for land-use types in species-level mosquito abundance models.** Posterior mean estimates, lower (2.5%) and upper (97.5%) credible intervals (CI) for land-use types in abundance models of four *Aedes* and four *Anopheles* mosquitoes.
(DOCX)

**S15 Table. Sites by land-use type and species included in the dataset.** Number of unique sites by land-use type where *Aedes* and *Anopheles* mosquito species were recorded.
(DOCX)

**S1 Fig. PRISMA flow of the systematic data search process.** PRISMA flow diagram of the systematic data collection process for mosquito biodiversity data in Latin America and the Caribbean. Three databases were searched (Medline, Scopus and Web of Science) and results combined before studies were screened by title and abstract (n = 8,554). A total of 1,790 studies were screened by full text, leading to inclusion of 85 studies that had suitable data.
(TIF)

**S2 Fig. Observed and fitted observations for models of mosquito abundance and species richness.** Observed and fitted model A) abundance (log +1) and B) species richness in models of total and *Aedes* and *Anopheles* mosquitoes. Red line represents the expectation if observed values equal fitted values.
(TIF)

**S3 Fig. Distribution of studies included in the mosquito biodiversity dataset by country.** Number of included studies by country in Latin America and the Caribbean. The total number of included studies was 93. Base map sourced from rnaturalearth (50).
(TIF)

**S4 Fig. Responses of total *Aedes* and *Anopheles* mosquito richness and abundance to land-use type and intensity.** Total (*Aedes* and *Anopheles*) mosquito richness (A) and abundance (B) responses to land-use types with minimal (circles), substantial (triangles) and combined (squares) use intensities. Effect sizes were adjusted to a percentage by expressing each mean fixed effect and 95% credible intervals as a percentage of the baseline (primary vegetation minimal use, shown as zero). Intensity levels for secondary vegetation, managed and urban land uses were aggregated due to a lack of data representation.
(TIF)

**S5 Fig. Geographical cross-validation of genus-level abundance and richness responses to land-use type and intensity.** Response of *Aedes* (A, C) and *Anopheles* (B, D) mosquitoes to land-use type and intensity excluding sites from Brazil. Dark grey estimates show the genus-level richness (A-B) and abundance (C-D) models with all the data and the light grey estimates show modelled estimates excluding sites from Brazil. Effect sizes were adjusted to a percentage by expressing each mean fixed effect and 95% credible intervals as a percentage of the baseline (primary vegetation minimal use, shown as zero). Intensity levels for secondary vegetation, managed and urban land uses were aggregated due to a lack of data representation.
(TIF)

**S6 Fig. Ecoregion sensitivity analysis.** Response of *Aedes* (A, C) and *Anopheles* (B, D) mosquito species richness (A-B) and abundance (C-D) to land-use type and intensity excluding each ecoregion in turn. Colours represent each ecoregion that was excluded. Effect sizes were adjusted to a percentage by expressing each mean fixed effect and 95% credible intervals as a percentage of the baseline (primary vegetation minimal use, shown as zero). Intensity levels for secondary vegetation, managed and urban land uses were aggregated due to a lack of data representation. Both abundance and species richness were highly sensitive to rainforest sites (pink—tropical and subtropical moist broadleaf forests).
(TIF)

**S7 Fig. Random subsampling cross-validation analysis.** Response of *Aedes* (A, C) and *Anopheles* (B, D) mosquito species richness (A-B) and abundance (C-D) to land-use type and intensity excluding 12.5% of the data at time. Colours represent each data group. Effect sizes were adjusted to a percentage by expressing each mean fixed effect and 95% credible intervals as a percentage of the baseline (primary vegetation minimal use, shown as zero). Intensity levels for secondary vegetation, managed and urban land uses were aggregated due to a lack of data representation.
(TIF)

**S8 Fig. Species-level cross-validation of genus-level abundance responses to land-use type and intensity.** Response of *Aedes* (A) and *Anopheles* (B) mosquito abundance to land-use type and intensity excluding influential species. Dark grey estimates show the genus-level

abundance model with all the data and the light grey estimates show modelled estimates excluding data for each species. For each genus, the four most represented species in the dataset were selected. Effect sizes were adjusted to a percentage by expressing each mean fixed effect and 95% credible intervals as a percentage of the baseline (primary vegetation minimal use, shown as zero). Intensity levels for secondary vegetation, managed and urban land uses were aggregated due to a lack of data representation.
(TIF)

## Acknowledgments

Special thanks to members of the Planetary Health & Infectious Disease (PHID) Lab at for their support and the Kate Jones' lab at UCL for informal feedback on this work.

## Author Contributions

**Conceptualization:** Isabel K. Fletcher, Rachel Lowe, Kate E. Jones.

**Data curation:** Isabel K. Fletcher, Rory Gibb.

**Formal analysis:** Isabel K. Fletcher.

**Funding acquisition:** Isabel K. Fletcher.

**Investigation:** Isabel K. Fletcher.

**Methodology:** Isabel K. Fletcher, Rory Gibb, Kate E. Jones.

**Resources:** Rachel Lowe, Kate E. Jones.

**Software:** Isabel K. Fletcher, Rory Gibb.

**Supervision:** Rachel Lowe, Kate E. Jones.

**Validation:** Isabel K. Fletcher, Rory Gibb.

**Visualization:** Isabel K. Fletcher.

**Writing – original draft:** Isabel K. Fletcher.

**Writing – review & editing:** Rory Gibb, Rachel Lowe, Kate E. Jones.

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
