## [Decision Letter · Decision Letter 0]

1 Mar 2023

Dear Dr Fletcher,

Thank you very much for submitting your manuscript "Differing taxonomic responses of mosquito vectors to anthropogenic land-use change in Latin America and the Caribbean" for consideration at PLOS Neglected Tropical Diseases. As with all papers reviewed by the journal, your manuscript was reviewed by members of the editorial board and by several independent reviewers. The reviewers appreciated the attention to an important topic. Based on the reviews, we are likely to accept this manuscript for publication, providing that you modify the manuscript according to the review recommendations. 

All three reviewers felt that the general topic was interesting and the analytical approach generally appropriate. However reviewer two felt the central question posed by the introduction (do mosquitos differ in there response to land use?) relatively trivial. I would not say that I found the central premise too well studied to be inherently interesting. that said, I also I agree with reviewer two that the unintuitive results with respect to pace of deforestation and changes in abundance is intriguing and worth highlighting more. I did not pick up on this at all on a quick readthrough, and I imagine many readers will also miss it. I leave it to your discretion how to better emphasize this result. However, I do not think the paper needs to be completely reframed around this topic, nor would I recommend adding any additional analyses, since the other two reviewers found the paper overall quite strong. In addition to this issue, all three reviewers suggest a host of smaller revisions all of which should be addressed in your response letter.

Sincerely,

Patrick R Stephens, Ph.D.

Guest Editor

Dileepa Ediriweera

Section Editor

All three reviewers felt that the general topic was interesting and the analytical approach generally appropriate. However reviewer two felt the central question posed by the introduction (do mosquitos differ in there response to land use?) relatively trivial. I would not say that I found the central premise too well studied to be inherently interesting. that said, I also I agree with reviewer two that the unintuitive results with respect to pace of deforestation and changes in abundance is intriguing and worth highlighting more. I did not pick up on this at all on a quick readthrough, and I imagine many readers will also miss it. I leave it to your discretion how to better emphasize this result. However, I do not think the paper needs to be completely reframed around this topic, nor would I recommend adding any additional analyses, since the other two reviewers found the paper overall quite strong. In addition to this issue, all three reviewers suggest a host of smaller revisions all of which should be addressed in your response letter.

Reviewer's Responses to Questions

**Key Review Criteria Required for Acceptance?**

**Methods**

-Are the objectives of the study clearly articulated with a clear testable hypothesis stated?

-Is the study design appropriate to address the stated objectives?

-Is the population clearly described and appropriate for the hypothesis being tested?

-Is the sample size sufficient to ensure adequate power to address the hypothesis being tested?

-Were correct statistical analysis used to support conclusions?

-Are there concerns about ethical or regulatory requirements being met?

Reviewer #1: The study was well defined from the start. Although the authors do not specifically focus on diseases, they are building a nice rationale about the role of mosquito vectors in transmitting zoonotic pathogens. Sample size is more than sufficient, and the hypothesis and predictions were well stated.

Reviewer #2: (No Response)

Reviewer #3: Yes to all questions. Suggestions are noted in the file attached.

**Results**

-Does the analysis presented match the analysis plan?

-Are the results clearly and completely presented?

-Are the figures (Tables, Images) of sufficient quality for clarity?

Reviewer #1: As far as I can tell, the statistical analyses have been done prperly. The authors used Bayesian Multi-level models to infer the role of land-use on mosquito species richness and abundance. Baseline was clearly define and same appropriate for their study. All the analyses seem to have been interpreted correctly, and the results and their directionality are well presented.

Reviewer #2: (No Response)

Reviewer #3: Yes to all questions. Suggestions are noted in the file attached.

**Conclusions**

-Are the conclusions supported by the data presented?

-Are the limitations of analysis clearly described?

-Do the authors discuss how these data can be helpful to advance our understanding of the topic under study?

-Is public health relevance addressed?

Reviewer #1: The conclusions of this study are appropriate, and limitations have been well defined. The authors put their study into perspective with other related work, and finish their discussion by discussing the impact of their study in the global disease landscape.

Reviewer #2: (No Response)

Reviewer #3: Yes to all questions. Suggestions are noted in the file attached.

**Editorial and Data Presentation Modifications?**

Reviewer #1: Minor comments:

Line 57: Global warming?

Line 71: I am not sure this statement is true; I can think of a few studies that were published in the past few years specifically focused on large-scale mosquito patterns (see Steiger et al. 2016, Rakotoarinia et al. 2022), and even more so on the amazon basin (MacDonald and Mordecal 2019). 

Line 78: I would argue that it negatively impacts native biodiversity, not biodiversity as a whole.

Line 150: The authors mention that mosquito abundance was corrected for sampling effort when it was reported. What happened with the data for the studies that did not gave sampling effort, were they excluded? If not, how were they integrated? Perhaps using a negative binomial distribution with a zero-inflated correction would be better.

Figure 3: I do not believe an abundance in the negative is possible, please change scale

Reviewer #2: (No Response)

Reviewer #3: Yes to all questions. Suggestions are noted in the file attached. Minor revisions.

**Summary and General Comments**

Reviewer #1: (No Response)

Reviewer #2: (No Response)

Reviewer #3: Suggestions are noted in the file attached.

PLOS authors have the option to publish the peer review history of their article (what does this mean?). If published, this will include your full peer review and any attached files.

Reviewer #1: Yes: Antoine Filion

Reviewer #2: No

Reviewer #3: No

Figure Files:

Data Requirements:

Reproducibility:

References

---

## [Decision Letter · Decision Letter 1]

30 May 2023

Dear Dr Fletcher,

Thank you very much for submitting your manuscript "Differing taxonomic responses of mosquito vectors to anthropogenic land-use change in Latin America and the Caribbean" for consideration at PLOS Neglected Tropical Diseases. As with all papers reviewed by the journal, your manuscript was reviewed by members of the editorial board and by several independent reviewers. The reviewers appreciated the attention to an important topic. Based on the reviews, we are likely to accept this manuscript for publication, providing that you modify the manuscript according to the review recommendations. 

The reviewers all found the manuscript to be greatly improved. However, reviewers two and three founds some additional minor issues that should be addressed before the manuscript is is finalized. Reviewer two felt that a minor detail of the methods could use additional clarification, and reviewer three found some additional typos which are detailed in a commented draft of your manuscript. Once you address these small remaining issues, this should be ready to go. I do not anticipate needing to send this out for a third round of review.

Sincerely,

Patrick R Stephens, Ph.D.

Guest Editor

Dileepa Ediriweera

Section Editor

The reviewers all found the manuscript to be greatly improved. However, reviewers two and three founds some additional minor issues that should be addressed before the manuscript is is finalized. Reviewer two felt that a minor detail of the methods could use additional clarification, and reviewer three found some additional typos which are detailed in a commented draft of your manuscript. Once you address these small remaining issues, this should be ready to go. I do not anticipate needing to send this out for a third round of review.

Reviewer's Responses to Questions

**Key Review Criteria Required for Acceptance?**

**Methods**

-Are the objectives of the study clearly articulated with a clear testable hypothesis stated?

-Is the study design appropriate to address the stated objectives?

-Is the population clearly described and appropriate for the hypothesis being tested?

-Is the sample size sufficient to ensure adequate power to address the hypothesis being tested?

-Were correct statistical analysis used to support conclusions?

-Are there concerns about ethical or regulatory requirements being met?

Reviewer #1: The authors have adequately answered all my comments

Reviewer #2: L178-180 ‘The best-fitting random effects structure was selected by formulating iterative models of each response (i.e. abundance and richness) with the addition of each random effect (Table S7-8)’ & L183-184 ‘Random effects were retained in the final model structure if the model was improved.’

Why are random effects tested? As they are there to take into account the non-independence of the data, they should not be removed from the model. Are the results the same when all random effects are included (even those that do not improve the fit of the models)?

Reviewer #3: Are the objectives of the study clearly articulated with a clear testable hypothesis stated? YES

-Is the study design appropriate to address the stated objectives? YES

-Is the population clearly described and appropriate for the hypothesis being tested? YES

-Is the sample size sufficient to ensure adequate power to address the hypothesis being tested? YES

-Were correct statistical analysis used to support conclusions? YES

-Are there concerns about ethical or regulatory requirements being met? YES

**Results**

-Does the analysis presented match the analysis plan?

-Are the results clearly and completely presented?

-Are the figures (Tables, Images) of sufficient quality for clarity?

Reviewer #1: The authors have adequately answered all my comments

Reviewer #2: Yes to all questions

Reviewer #3: -Does the analysis presented match the analysis plan? YES

-Are the results clearly and completely presented? YES

-Are the figures (Tables, Images) of sufficient quality for clarity? YES

**Conclusions**

-Are the conclusions supported by the data presented?

-Are the limitations of analysis clearly described?

-Do the authors discuss how these data can be helpful to advance our understanding of the topic under study?

-Is public health relevance addressed?

Reviewer #1: The authors have adequately answered all my comments

Reviewer #2: Yes to all questions

Reviewer #3: -Are the conclusions supported by the data presented? YES

-Are the limitations of analysis clearly described? YES

-Do the authors discuss how these data can be helpful to advance our understanding of the topic under study? YES

-Is public health relevance addressed? YES

**Editorial and Data Presentation Modifications?**

Reviewer #1: The authors have adequately answered all my comments

Reviewer #2: No suggestion

Reviewer #3: THERE ARE FEW GRAMMATICAL ERRORS THAT NEED ATTENTION.

**Summary and General Comments**

Reviewer #1: The authors have adequately answered all my comments

Reviewer #2: The authors have done a good job revising the manuscript and responding to all comments. I have no additional comments to make other than my comment in the 'Methods' section.

Reviewer #3: WELL DONE ARTICLE, IT SHOULD BE ACCEPTED WITH MINOR REVISIONS.

PLOS authors have the option to publish the peer review history of their article (what does this mean?). If published, this will include your full peer review and any attached files.

Reviewer #1: No

Reviewer #2: No

Reviewer #3: No

Figure Files:

Data Requirements:

Reproducibility:

References

---

## [Editor Report · Decision Letter 2]

8 Jun 2023

Dear Dr Fletcher,

We are pleased to inform you that your manuscript 'Differing taxonomic responses of mosquito vectors to anthropogenic land-use change in Latin America and the Caribbean' has been provisionally accepted for publication in PLOS Neglected Tropical Diseases.

Best regards,

Patrick R Stephens, Ph.D.

Guest Editor

Dileepa Ediriweera

Section Editor

---

## [Editor Report · Acceptance letter]

30 Jun 2023

Dear Dr Fletcher,

We are delighted to inform you that your manuscript, "Differing taxonomic responses of mosquito vectors to anthropogenic land-use change in Latin America and the Caribbean," has been formally accepted for publication in PLOS Neglected Tropical Diseases.

Best regards,

Shaden Kamhawi

co-Editor-in-Chief

Paul Brindley

co-Editor-in-Chief
